# Poly (ADP-Ribose) Polymerase Inhibitor Olaparib-Resistant *BRCA1*-Mutant Ovarian Cancer Cells Demonstrate Differential Sensitivity to PARP Inhibitor Rechallenge

**DOI:** 10.3390/cells13221847

**Published:** 2024-11-07

**Authors:** Chi-Ting Shih, Tzu-Ting Huang, Jayakumar R. Nair, Kristen R. Ibanez, Jung-Min Lee

**Affiliations:** Women’s Malignancies Branch, Center for Cancer Research, National Cancer Institute, National Institutes of Health (NIH), Bethesda, MD 20892, USA; tzu-ting.huang@nih.gov (T.-T.H.); jayakumar.nair@nih.gov (J.R.N.); kr892361@ucf.edu (K.R.I.); leej6@mail.nih.gov (J.-M.L.)

**Keywords:** poly (ADP-ribose) polymerase inhibitor resistance, ovarian cancer, *BRCA*-mutation, replication fork, single-strand breaks

## Abstract

Poly (ADP-ribose) polymerase inhibitors (PARPis) show cytotoxicity in homologous recombination deficiency (HRD) seen in *BRCA*-mutant ovarian cancer (OvCa). Despite initial responses, resistance often develops. The reintroduction of different PARPis, such as niraparib or rucaparib, has shown some clinical activity in *BRCA* mutation-associated OvCa patients with prior olaparib treatment, yet the underlying mechanisms remain unclear. To investigate the differential sensitivity to different PARPis, we established an olaparib-resistant *BRCA1*-mutant OvCa cell line (UWB-OlaJR) by exposing UWB1.289 cells to gradually increasing concentrations of olaparib. UWB-OlaJR exhibited restored HR capability without *BRCA1* reversion mutation or increased drug efflux. We examined cell viability, DNA damage, and DNA replication fork dynamics in UWB-OlaJR treated with various PARPis. UWB-OlaJR exhibits varying sensitivity to PARPis, showing cross-resistance to veliparib and talazoparib, and sensitivity with increased cytotoxicity to niraparib and rucaparib. Indeed, DNA fiber assay reveals that niraparib and rucaparib cause higher replication stress than the others. Moreover, S1 nuclease fiber assay shows that niraparib and rucaparib induce greater DNA single-strand gaps than other PARPis, leading to increased DNA damage and cell death. Our study provides novel insights into differential PARPi sensitivity in olaparib-resistant *BRCA*-mutant OvCa, which requires further investigation of inter-agent differences in large prospective studies.

## 1. Introduction

Poly (ADP-ribose) polymerase 1 (PARP1) is an abundant nuclear protein that post-translationally attaches poly ADP-ribose (PAR) to both itself and other target proteins such as histones [1,2], chromatin remodelers (nuclear receptor binding SET domain protein 2, amplification in liver cancer 1) [3,4,5], transcription factors (Kruppel-like factor 4) [6] and DNA damage repair proteins (breast cancer type 1 susceptibility protein (BRCA1), DNA polymerase β (POLβ), X-ray repair cross-complementing protein 1 (XRCC1), and DNA ligase 3 (LIG3) [7,8]. Among the DNA damage repair pathways, poly ADP-ribosylation (PARylation) activity mainly contributes to single-strand break (SSB) repair processes, i.e., base excision repair and nucleotide excision repair, and double-strand break (DSB) repair response such as homologous recombination (HR), non-homologous end joining (NHEJ), and microhomology-mediated end joining (MMEJ) [9]. PARP1 also involves replication fork (RF) dynamics during DNA replication by recognizing stalled RFs and stabilizing via promoting fork reversal [10]. PARP1 also protects reversed RFs by inhibiting RecQ protein-like 1 (RECQ1) against meiotic recombination 11 (MRE11)-mediated nucleolytic degradation [11,12]. Additionally, PARP1 modulates chromatin structure and facilitates DNA repair through PARylating histones, which results in chromatin relaxation [13]. Moreover, PARP1 senses incompletely processed lagging strands and subsequently recruits XRCC1 and LIG3 to promote ligation [14,15]. Overall, these roles of PARP1 reflect its complex function in maintaining genome stability and effective DNA damage response.

The interaction of PARP inhibition in cancer cells with HR repair deficiency (HRD), i.e., ovarian and breast cancers with *BRCA* mutations, has been well-studied. Briefly, inhibition of PARP1 enzyme activity by PARP inhibitors (PARPis) leads to the accumulation of unrepaired SSBs, which are subsequently converted into DSBs during DNA replication, resulting in synthetic lethality and cell death in HRD cells [16,17]. Clinically, tumor shrinkages by PARPis, including olaparib, rucaparib, or niraparib, have been observed in both HRD and non-HRD high-grade serous ovarian cancer (HGSOC) patients, suggesting additional mechanisms may contribute to PARPi efficacy beyond DNA DSB repair pathways [18]. As such, the accumulation of ssDNA gaps but not DSBs has been recently shown to be a driver of cell death in *BRCA*-deficient cells [19]. Olaparib is also shown to disrupt Okazaki fragment processing (OFP) by blocking PARP1-mediated recruitment of XRCC1 and LIG3, inducing lagging strand gaps that impede the maturation of nascent DNA strands [15,20]. These findings suggest that PARPis involve various steps of the DNA repair process. Thus, more diverse preclinical models are needed to understand the secondary resistance to PARPis.

Currently, there are four PARPis—olaparib, rucaparib, niraparib, and talazoparib—approved by the United States Food and Drug Administration (FDA) for treatment or front-line maintenance therapy of ovarian cancer (OvCa) or breast cancer [21]. In OvCa, olaparib was the first FDA-approved PARPi that significantly improved clinical outcomes for patients with *BRCA*-mutant (*BRCA*m) HGSOC [22,23,24]. However, despite their efficacy, acquired resistance to PARPis is inevitable. The major mechanisms of resistance to PARPi include reversion mutations in HR repair genes (e.g., *BRCA1/2* and *RAD51*), stabilization of RFs, increased PARylation activity, and enhanced drug efflux [25]. These mechanisms often coexist in PARPi-resistant tumors [25]. Hence, further understanding of these resistance mechanisms is necessary to identify the potential of PARPi rechallenge, especially in the PARPi-resistant setting.

A few clinical studies have suggested that the reintroduction of PARPis to relapsed OvCa patients who received prior platinum-based therapy or PARPi may offer some clinical benefits. The phase III trial (OReO/ENGOT-ov38) demonstrated the benefit of olaparib rechallenge with a marginal yet statistically significant increase in median progression-free survival (PFS) over the placebo in both *BRCA*m patients (4.3 months vs. 2.8 months, hazard ratio = 0.57, 95% confidence interval (CI): 0.37–0.87; *p* = 0.022) and non-*BRCA*m patients (5.3 months vs. 2.8 months, hazard ratio = 0.43, 95% CI: 0.26–0.71; *p* = 0.0023) who had prior olaparib [26]. Similarly, Gauduchon et al. found improvements in median PFS when the same PARPi (niraparib: *n* = 24, olaparib: *n* = 45, and rucaparib: *n* = 5) was rechallenged in relapsed OvCa patients after local therapy such as radiation therapy and surgery [27]. Also, Moubarak et al. reported instances of extended PFS in re-treated patients using either the same or different PARPis, including olaparib, rucaparib, and niraparib [28]. These findings imply the potential benefits of second PARPi therapy in prior PARPi-exposed OvCa patients, but the mechanisms underlying these benefits remain elusive. 

To better understand the therapeutic potential of PARPi rechallenge in olaparib-resistant *BRCA1*-mutant (*BRAC1*m) OvCa cells, we explored their effects on cytotoxicity, replication dynamics, and DNA damage. Our results demonstrated that different PARPis exhibit varying degrees of sensitivity, with niraparib and rucaparib showing greater cytotoxicity compared to other PARPis. Mechanistically, niraparib and rucaparib decrease RF speed by inducing higher levels of SSBs, which is evidenced by the reduced length of DNA fibers after S1 nuclease treatment as well as by the increased replication protein A1 (RPA1) foci in olaparib-resistant *BRCA1*m HGSOC cells. Together, our study provides novel mechanistic insights into the reintroduction of niraparib and rucaparib for olaparib-resistant *BRCA1*m OvCa. These findings also provide supporting preclinical models for future clinical studies aimed at investigating inter-agent differences in the PARPi rechallenge of PARPi-resistant populations.

## 2. Materials and Methods

### 2.1. Drug Preparation

PARPis including olaparib (#S1060), rucaparib (#S4948), niraparib (#S2741), talazoparib (#S7048), and veliparib (#S1004) were purchased from Selleck Chemicals (Houston, TX, USA). Moreover, 100 mM drug stocks were prepared in dimethyl sulfoxide (DMSO; #S-002-M, Sigma-Aldrich, Saint Louis, MO, USA) and stored in aliquots at −80 °C until use. For clinical relevance, clinically attainable concentrations of each PARPi were used: 10–20 µM for olaparib, 5–10 µM for veliparib, rucaparib, niraparib, and 50–100 nM for talazoparib [29,30,31,32,33]. Hydroxyurea (HU; #H8627) was from Sigma-Aldrich and was used to block replication fork progression. Moreover, 2M of HU were prepared as stock in DMSO. Methyl methanesulfonate (MMS, #129925, Sigma-Aldrich) was dissolved directly into the culture medium. Emetine (#324693, Sigma-Aldrich) solution was prepared in PBS (2 mM). 

### 2.2. Development of PARPi Olaparib-Resistant Cell Lines

A PARPi-resistant *BRCA1*m HGSOC cell line, UWB-OlaJR, was established in-house from a *BRCA1*-null UWB1.289 (*BRCA1* 2594delC). The parental UWB1.289 (#CRL-2945) cell line was purchased from ATCC (Manassas, VA, USA) and cultured by gradually increasing concentrations of olaparib from 0.4 µM to 20 µM over 10 months. Resistance of UWB-OlaJR to olaparib was confirmed by a 5-day XTT assay [34]. UWB1.289 cells were maintained in RPMI1640 medium with (+) L-glutamine supplemented with 10% fetal bovine serum, 0.01 mg/mL insulin, and 1% penicillin/streptomycin. UWB-OlaJR cells were routinely maintained at 20 µM olaparib. Cells were cultured without olaparib for at least 3 days before being used for subsequent experiments. Authentication was evaluated by short tandem repeat analysis conducted by ATCC and was confirmed negative for mycoplasma using MycoAlert (#NC9719283, Lonza, Rockville, MD, USA).

### 2.3. Whole Exome Sequencing (WES)

WES was performed to identify the genomic alterations of UWB1.289 and its counterpart, UWB-OlaJR. Genomic DNAs from UWB1.289 and UWB-OlaJR were prepared using QIAamp DNA Micro Kit (#56304, Qiagen, Germantown, MD, USA). Exome samples were pooled and sequenced on NovaSeq run using Agilent SureSelect XT (Agilent, Santa Clara, CA, USA) and paired-end sequencing mode at the Center for Cancer Research sequencing facility, National Cancer Institute, as described before [35].

### 2.4. Sanger Sequencing

The sequencing of *BRCA1* exon 11 was performed using the Sanger method. Genomic DNA was extracted from both UWB1.289 and UWB-OlaJR using QIAamp DNA Micro Kit (#56304, Qiagen). The target DNA regions were amplified by polymerase chain reaction (PCR) using specific primers tagged with an M13 tag. The forward primer was ordered as “5′-GGAAACAGCTATGACCATGCCAACCACAGGAAAGCCTGC-3′” and the reverse primer was ordered as “5′-GTAAAACGACGGCCAGTCCTGGTACTGATTATGGCACTCAGGA-3′” (Integrated DNA Technologies, Coralville, IA, USA) and amplified with Platinum™ SuperFi™ PCR Master Mix (12.5 µL; Thermo Fisher Scientific, Waltham, MA, USA) with kit-provided GC Enhancer (5 µL). Amplification was performed on Applied Biosystems’ ProFlex PCR System (Thermo Fisher Scientific) using PCR conditions: 98 °C for 30 s, 35 cycles of 98 °C for 10 s, 65 °C for 10 s, and 72 °C for 30 s, followed by a hold at 72 °C for 5 min and then an infinite hold at 4 °C. The resulting products were then checked for quality and concentration with Agilent’s 2100 Bioanalyzer (Agilent Technologies, Santa Clara, CA, USA) and Agilent’s DNA 1000 kit, using Bioanalyzer software version 2100 Expert B.02.11 SI824. Samples were diluted to 1.8 ng/µL, then purified using exonuclease I (GE Healthcare, Pittsburgh, PA, USA) and shrimp alkaline phosphatase (SAP; Affymetrix USB, Thermo Fisher Scientific), in accordance with the Exo-Sap protocol. The Exo-Sap-sample mixture was then incubated in the ProFlex™ PCR System: 37 °C for 15 min, then 80 °C for 15 min, followed by a 4 °C hold. This purified amplicon then proceeded into cycle sequencing with Applied Biosystems’ BigDye™ Terminator v3.1 Cycle Sequencing Kit (Thermo Fisher Scientific) and M13 Forward and M13 Reverse primers (Invitrogen, Waltham, MA, USA), using the following conditions in the ProFlex™ PCR System: 96 °C for 1 min, 25 cycles of 96 °C for 10 s, 50 °C for 5 s, 60 °C for 1 min and 15 s; followed by a hold at 4 °C. Samples were then purified using Mag-Bind SeqDTR (Omega Bio-Tek, Norcross, GA, USA), in accordance with the Mag-bind SeqDTR protocol and then processed on Applied Biosystems’ 3730xl DNA Analyzer (Thermo Fisher Scientific), 96 capillary 50 cm array, using data generated with 3730/3730XL DNA Analyzer Sequencing Standards, BigDye™ Terminator v3.1 Kit (Thermo Fisher Scientific) and 3730XL Data Collection Software (version 5.0; Thermo Fisher Scientific). Data were then analyzed using Mutation Surveyor (version 5.1.2; SoftGenetics, State College, PA, USA).

### 2.5. Cell Growth Assay

Three thousand cells/well were seeded in 96-well plates and treated with indicated drugs or 0.01% DMSO as control after seeding. After being treated with indicated drugs for 5 days, cell viability was assessed by the XTT assay (#X6493, Thermo Fisher Scientific, Pittsburg, PA, USA) according to the manufacturer’s instructions. The absorbance was measured by Synergy™ HTX Multi-Mode Microplate Reader with Gen5™ software version 2.0 (BioTek Instruments, Winooski, VT, USA). IC_50_ values were calculated using GraphPad Prism version 10.2.0 (GraphPad Software, Boston, MA, USA). For DNA SSB induction experiments, cells were incubated with 0.01% MMS for 10 min and then replaced with RPMI medium containing indicated concentrations of PARPis. For the replication inhibition experiments, cells were incubated with 10 µM of emetine for 30 min and then replaced with RPMI medium containing indicated concentrations of PARPis. 

### 2.6. Clonogenic Assay 

Ten thousand cells were seeded in 12-well plates and treated with PARPis after seeding. After 10 days, the media were aspirated, and colonies were fixed in methanol and stained with 0.01% crystal violet solution. Colony images were scanned, then quantified by colony area percentage to reflect cell survival using a ColonyArea Image J Plugin [36] (ImageJ version 1.54d, NIH, Bethesda, MD, USA).

### 2.7. Immunofluorescence (IF) Staining

BRCA1, γH2AX, RAD51, and RPA1 foci formation were examined by IF staining. The cells were grown on ibidiTM 8-well removable chamber slides (ibidi USA Inc., Fitchburg, WI, USA) and incubated with each PARPi. For the evaluation of γH2AX and RAD51 foci, the cells were incubated for 48 h, whereas the cells were incubated for 6 h for RPA1 foci. Cells were fixed afterward in 4% paraformaldehyde for 10 min, permeabilized with 0.5% Triton-X 100 for another 10 min, then blocked with 5% bovine serum albumin for 1 h at 37 °C. The slides were incubated overnight with primary antibodies followed by appropriate secondary antibodies for 1 h at room temperature. All slides were mounted with VECTASHIELD HardSet Antifade Mounting Medium with DAPI (#H-1500, Vector Laboratories, Burlingame, CA, USA). Antibodies used for IF staining are provided in Table 1. Images were collected with a Nikon SoRa Spinning Disk confocal microscope (Tokyo, Japan) with a 60×/1.49 oil immersion objective. BRCA1, γH2AX, RAD51, and RPA1 were quantified using ImageJ to evaluate the number of foci per nuclei, and all cells with >5 foci were considered positive cells.

### 2.8. Multidrug Resistance (MDR) Activity Assay

Cells were incubated with DMSO, positive controls (cyclosporin A [1:1000]; verapamil [1:2000]), and PARPis for 30 min at 37 °C. Drug efflux activity and the influence of different treatments in UWB1.289 and UWB-OlaJR were measured using the Cayman Multidrug Resistance Assay kit (#600370, Cayman Chemical, Ann Arbor, MI, USA) according to the manufacturer’s instructions. Intracellular calceinAM fluorescence was measured by SynergyTM HTX Multi-Mode Microplate Reader with Gen5TM software version 2.0 (BioTek Instruments).

### 2.9. Neutral Comet Assay

Neutral comet assay was conducted to examine the DSBs as previously described [37]. After the electrophoresis on day 1, the slides were stained with SYBR Green dye for 5 min in the dark on day 2. Slides were then washed 3 times with deionized water and allowed to dry in the dark overnight. At least 100 cells were scored in each experiment. The mean tail moment was calculated as an index of DNA damage using CometScore Pro (TriTek Corporation, Sumerduck, VA, USA).

### 2.10. DNA Fiber Assay

DNA fiber assays were conducted to assess RF speed, protection, and restart. Cells were treated with clinically attainable concentrations of each PARPi [30,31,32,33,34]. To measure replication fork speed, cells were labeled with 60 µM 5-chloro-2′-deoxyuridine (CldU; #C6891, Sigma-Aldrich) and 500 µM 5-Iodo-2′-deoxyuridine (IdU; #I7125, Sigma-Aldrich) for 30 min each with or without PARPis. Replicating 1 µm roughly corresponds to 2.59 kb [38]. For RF protection, cells were labeled with CldU and IdU for 20 min each, followed by treatment with or without PARPis or 2 mM of HU for another 3 h. For the S1 nuclease fiber assay, cells were treated with PARPis for 2 h, followed by 20 min of CldU pulse. Cells were then treated with CSK100 buffer for 5 min at room temperature and incubated with or without 20 U/mL S1 nuclease (#18001-016, Invitrogen, for 30 min at 37 °C. Cells were scraped, collected in PBS + 0.1% BSA, and lysed with lysis buffer (0.5% sodium dodecyl sulfate, 200 mM Tris-HCl [pH 7.4], 50 mM EDTA). Labeled DNAs with CldU and IdU were stained with mouse anti-IdU primary antibody (1:250; #NBP2-44056, Novus Biological, Centennial, CO, USA) and rat anti-CldU primary (1:200; #NB500-169, Novus Biological), respectively. Anti-rat Alexa Fluor 488 (1:250; #A-11006, Thermo Fisher Scientific) and anti-mouse Alexa 594 (1:250; #A-11005, Thermo Fisher Scientific) were used for secondary antibodies. Images were captured with a Zeiss LSM 780 confocal microscope (Oberkochen, Baden-Württemberg, Germany). Fiber length was measured using ImageJ software version 1.54d. At least 50 fibers are quantified for each experiment.

### 2.11. Statistical Analysis

All statistical analyses were performed and figured in GraphPad Prism version 10.2.0 (GraphPad Software, Boston, MA, USA). All data were repeated in triplicate and analyzed using Student’s *t*-test or one-way ANOVA. Data were shown as mean ± SD, and *p* < 0.05 was considered statistically significant.

## 3. Results

### 3.1. Establishment of BRCA1m Olaparib-Resistant OvCa Cell Line

To establish an olaparib-resistant UWB-OlaJR cell line, we gradually increased the exposure of UWB1.289 cells to higher olaparib concentrations (0.4–20 μM) over a 10-month period (Figure 1A). No distinct morphological changes were observed between the two cell lines (Figure 1B). As demonstrated by a 5-day XTT assay, UWB-OlaJR cells (IC_50_: 33.78 μM) were 47-fold more resistant to olaparib compared to the parental line (IC_50_: 0.72 µM, *p* = 0.0005; Figure 1C). Similarly, UWB-OlaJR showed greater resistance to olaparib relative to parental cells evaluated by a 10-day colony-forming assay (Figure 1D) and maintained stable PARPi resistance after 8 weeks of olaparib withdrawal (Figure 1E). UWB-OlaJR cells were also cross-resistant to cisplatin (Figure 1F), consistent with previous reports in PARPi-resistant OvCa cells [39,40].

### 3.2. UWB-OlaJR Shows a Varying Degree of Sensitivity to Different PARPis

Next, we evaluated the cross-resistance of UWB-OlaJR cells with four different PARPis. We selected four PARPis because rucaparib and niraparib have received FDA approval for OvCa, while talazoparib has been approved for use in breast cancer. Veliparib was also included because of its antitumor activity and acceptable safety/tolerability in early-phase clinical trials [41]. UWB-OlaJR showed cross-resistance with other PARPis, as evidenced by increased IC_50_ values compared to its parental line (Figure 2A, veliparib: 7.4-fold, talazoparib: 1494.7-fold, niraparib: 128.4-fold, rucaparib: 6.2-fold). Although UWB-OlaJR exhibited resistance to all four PARPis at their clinically attainable concentrations (indicated by dotted lines on each graph), more than 50% of the cells survived the highest concentration of veliparib and talazoparib in 5-day XTT assay (Figure 2A) and 10-day colony-forming assays (Figure 2B), in contrast to niraparib and rucaparib. We speculated differential cytotoxicity among various PARPis was unlikely associated with PARP trapping ability (Figure 2A,B) as both high PARP-trapper talazoparib and low PARP-trapper veliparib showed similar cytotoxicity on UWB-OlaJR [42,43]. We, therefore, explored other mechanisms of action contributing to the differential cytotoxicity.

### 3.3. A Restored DNA DSB Repair Function in Olaparib-Resistant UWB-OlaJR

We investigated whether increased DNA damage was associated with the differential cytotoxicity of PARPis. All tested PARPis induced a significant increase of γH2AX foci level, a DSBs marker, on PARPi-sensitive UWB1.289 cells (all *p* < 0.05, Figure 3A). Similarly, all PARPis increased the signal of DSBs marker in olaparib-resistant UWB-OlaJR cells (all *p* < 0.05, Figure 3A). Neutral comet assay also confirmed increased tail moments in both cell lines treated with PARPis (Figure 3B). Notably, UWB-OlaJR showed overall lower tail moments compared to the parental line when treated with PARPis (Figure 3B).

To determine if decreased DNA damage in UWB-OlaJR was driven by restored HR repair function, we assessed RAD51 foci, a marker of HR capability [44]. As expected, no RAD51 foci were observed in *BRCA1*-deficient UWB1.289 cells. In contrast, a significantly higher percentage of RAD51 foci-positive cells was seen in UWB-OlaJR cells (Figure 3C). However, no significant difference was found among the PARPis tested (Figure 3C), suggesting HR activity is unlikely a main factor for the differential sensitivity to PARPis in UWB-OlaJR. We also characterized *BRCA1* mutations in UWB-OlaJR by WES and Sanger sequencing of *BRCA1*, given that reversion mutations of *BRCA1* are known to restore BRCA1 functionality and confer resistance to PARPi [45]. WES showed no new *BRCA1* mutations in UWB-OlaJR compared to its parental line (Appendix A). Sanger sequencing of exon 11 that contains 2594delC also confirmed the mutation status of *BRCA1* in UWB-OlaJR remains the same as its parental cells (Appendix A). Lastly, UWB-OlaJR did not present BRCA1 foci when treated with olaparib (Appendix A), indicating UWB-OlaJR confers olaparib resistance by mechanisms other than restoring functional BRCA1 for HR repair reactivation. 

### 3.4. No Changes in Drug Efflux Activity in UWB-OlaJR Treated with Different PARPis

Different PARPis exhibit varying susceptibility to drug efflux pumps, which may impact intracellular concentrations and therapeutic efficacy [46]. Therefore, we examined their activity in cellular drug retention using calcein AM assay. Calcein AM is a cell-permeable non-fluorescent dye that is cleaved intracellularly to produce the impermeable fluorescent calcein, and we can assess the activity of drug efflux pumps, such as the ABCB1 transporter by measuring calcein retention [47]. Our results showed no significant difference in the retention levels of calcein in UWB-OlaJR cells treated with different PARPis compared to those treated with ABCB1 inhibitors (cyclosporin A and verapamil), which dramatically inhibited calcein efflux (Appendix A). These results suggest that drug efflux is unlikely to contribute to the varying cytotoxicity of PARPis in UWB-OlaJR.

### 3.5. Each PARPi Generates Varying Degrees of Replication Stress

Persistent RF instability increases DNA damage [20] and exacerbates replication stress, further sensitizing cells to PARPis [25]. We performed DNA fiber assays to study the impact of different PARPis on RF dynamics. We found that RF progression was significantly slower in both UWB1.289 and UWB-OlaJR after PARPis treatment, as evidenced by decreased RF speed compared to the untreated group (Figure 4A). Notably, rucaparib and niraparib stalled RF progression to a greater extent than other PARPis (Figure 4A), indicating that these two PARPis might interfere with RF stability or restart, leading to slower RF progression. However, nascent DNA degradation occurred after PARPis treatment in parental cells but not in UWB-OlaJR cells (Figure 4B), suggesting RF protection may not affect the differential sensitivity to PARPis.

### 3.6. Niraparib and Rucaparib Induce Higher Levels of SSBs That Induce Cell Death

Decreased RF speed may also result from unrepaired ssDNA lesions [48]. S1 nuclease digests and shortens the DNA fiber when nascent ssDNA breaks are within the labeled replication tracts [49]. To determine the connection between decreased RF speed and ssDNA gaps, we performed the DNA fiber assays with S1 nuclease. Treatment with S1 nuclease significantly reduced replication tract lengths in both parental and UWB-OlaJR cells compared to the control group (*p* < 0.0001, Figure 5A). Niraparib- and rucaparib-treated cells exhibited higher numbers of SSBs compared to the other three PARPis in both cell lines (Figure 5A). To identify whether the ssDNA gap is associated with PARPi response, we performed IF staining of ssDNA binding protein RPA1 foci [50]. The levels of RPA1 foci increased in all five PARPis-treated UWB1.289 cells but only in the niraparib- and rucaparib-treated UWB-OlaJR cells (Figure 5B), suggesting that these two PARPis may induce greater amounts of ssDNA gaps that lead to higher cytotoxicity.

Lastly, we treated the cells with PARPis in the presence of MMS [51], an alkylating agent that induces ssDNA gaps, to identify whether ssDNA gap induction would enhance PARPi sensitivity. MMS significantly sensitized UWB-OlaJR to olaparib, veliparib, and talazoparib (all *p* < 0.001, olaparib: 49.78-fold, veliparib: 103.23-fold, talazoparib: 288.5-fold) and further promoted the cytotoxicity of niraparib and rucaparib (all *p* < 0.05, niraparib: 3.96-fold, rucaparib: 8.22-fold) as shown by the decreased IC_50_ values (Figure 5C). Then, cells were treated with emetine, a DNA replication inhibitor [15], to evaluate whether emetine could mitigate the induction of replication-related ssDNA gaps. Blocking DNA replication rescued the cytotoxicity of niraparib and rucaparib (Appendix A). These results suggest that SSB inducers, such as MMS, can reverse PARPi resistance by creating more ssDNA gaps. Conversely, blocking replication reduces replicative ssDNA gaps, which may abolish PARPi cytotoxicity. 

In summary, among all five PARPis that have been tested, niraparib and rucaparib demonstrated greater cytotoxicity and reduced RF speed compared to the other three PARPis, likely due to the higher levels of SSB induction. Table 2 summarizes the different mechanisms of action of five PARPis when rechallenged in the PARPi olaparib-resistant *BRCA1*m HSGOC cells.

## 4. Discussion

There has been growing interest in the potential benefits of administering a second PARPi to patients who have previously benefited from PARPis [26]. Currently, the decision-making is primarily based on the expected side effects profiles rather than efficacy, as all PARPis have shown similar overall response rates (ORR) and PFS in the upfront maintenance therapy setting for *BRCA*m OvCa patients [52]. However, once patients progress after completing PARPi maintenance therapy and rechallenge is considered, it is unknown how to select the PARPi with the greatest potential benefit. While some clinical data have shown PARPi rechallenge may be effective in various contexts, these studies have not directly compared multiple PARPis, used relatively small patient cohorts, and the reasons behind these observations remain largely unexplored [26,53,54]. 

Here, we established an olaparib-resistant *BRCA1*m OvCa cell line to systematically study the differential responsiveness to the second PARPi treatment. We found that olaparib-resistant *BRCA1*m OvCa cells show greater sensitivity to niraparib and rucaparib relative to other PARPis. This effect was largely attributed to the various levels of induction of SSB and replication-related ssDNA gaps in the PARPi rechallenge setting. It is also noteworthy that HR or fork protection (FP) was not a main driver of PARPis’ differential cytotoxicity. Our findings align with the recent findings of PARP1 involving SSB repair and backup OFP [15,55], suggesting that different PARPi may induce varying degrees of SSB accumulation during the S phase, thus impeding RFs and leading to consequent cell death. 

Conventionally, BRCAness, which refers to HR and FP defects, is thought to render *BRCA*-deficient OvCa cells sensitive to PARPis due to their inability to prevent and repair DSBs. While this DSB-centric model has guided the development and clinical use of PARPi for decades, emerging evidence suggests that PARPi-induced cytotoxicity involves mechanisms beyond DSB induction [15,56]. Moreover, it is evident that there are other factors predicting the response to PARPi treatment besides HR and FP status in tumors [57]. Specifically, Panzarino et al. reported that the presence of persistent ssDNA replication gaps, rather than FP or HR defects, predicted the response of BRCA1-deficient triple-negative patient-derived breast cancer cells to HU [58]. Similarly, in PARPi-naïve *BRCA1*m OvCa cells, high levels of ssDNA gaps were associated with increased PARPi sensitivity. Notably, replication gaps also correlated with sensitivity to PARPi even in HR- and FP-proficient cells, thus implicating gaps as the sensitizing lesions [57]. Collectively, these findings, along with our results, indicate that ssDNA gaps, in addition to DSBs, are an important source of inducing PARPi sensitivity [18,57,58]. In addition, we demonstrate that ssDNA gaps may also confer sensitivity to another PARPi after the development of resistance to the first PARPi exposure. 

Recent studies on PARP1’s role in initiating OFP and filling unattended gaps on the lagging strand have uncovered a number of proteins that potentially affect PARPi cytotoxicity [59]. For instance, deficiencies in canonical OFP factors (i.e., exonuclease 1, DNA replication helicase/nuclease 2, LIG1, or proliferating cell nuclear antigen) as well as defects in proteins involved in backup OFP (i.e., XRCC1, LIG3, or POLβ) may increase the sensitivity of *BRCA2*-deficient cells to PARPis via inducing replication gaps [56,60,61]. Accordingly, restoration of backup OFP in BRCA1-deficient breast cancer cells suppresses gap formation via the loss of 53BP1 [57,62]. Furthermore, Cong et al. recently reported that loss of Fanconi anemia complementation group J (FANCJ) helicase—involved in replisome assembly and induction of PARP1 sequestering—results in the loss of PARP1 activity and subsequent development of olaparib resistance via blocking ssDNA gap induction [63]. The authors suggest that diminished PARP1 activity, rather than canonical PARP trapping, drives cytotoxicity in *BRCA*-deficient cells. Further investigation may prove useful in determining how different PARPis affect lagging strand synthesis or gap suppression mechanisms. 

There are limitations to this study. We acknowledge that additional olaparib-resistant *BRCA*m OvCa cell lines would help validate ssDNA gaps as a common resistance mechanism against PARPi in OvCa. Also, comparative analyses of ssDNA gap repair mechanisms and OFP pathway dynamics between UWB-OlaJR cells and other PARPi-resistant lines may uncover the compensatory DNA repair pathways for acquired resistance to PARPi. Moreover, the effects of other PARPis, specifically rucaparib and niraparib, on ssDNA gap formation and the backup OFP pathway require further investigation in various PARPi-resistant OvCa preclinical models. Ultimately, clinical studies testing different PARPi reintroduction strategies will provide potential therapeutic implications in OvCa patients.

## 5. Conclusions

In summary, we highlight the important role of ssDNA gaps as a determinant of cytotoxicity with PARPi rechallenge. Among the five PARPis we examined, niraparib and rucaparib demonstrated the most potent activity in inducing ssDNA gaps. Our findings may benefit the development of PARPi rechallenging strategies for patients with acquired resistance to PARPis. Also, it is worth considering the investigation of these mechanisms when developing synthetic lethality combinations with PARPis and identifying potential patients that might benefit from PARPi-based therapy.

## Figures and Tables

**Figure 1 cells-13-01847-f001:**
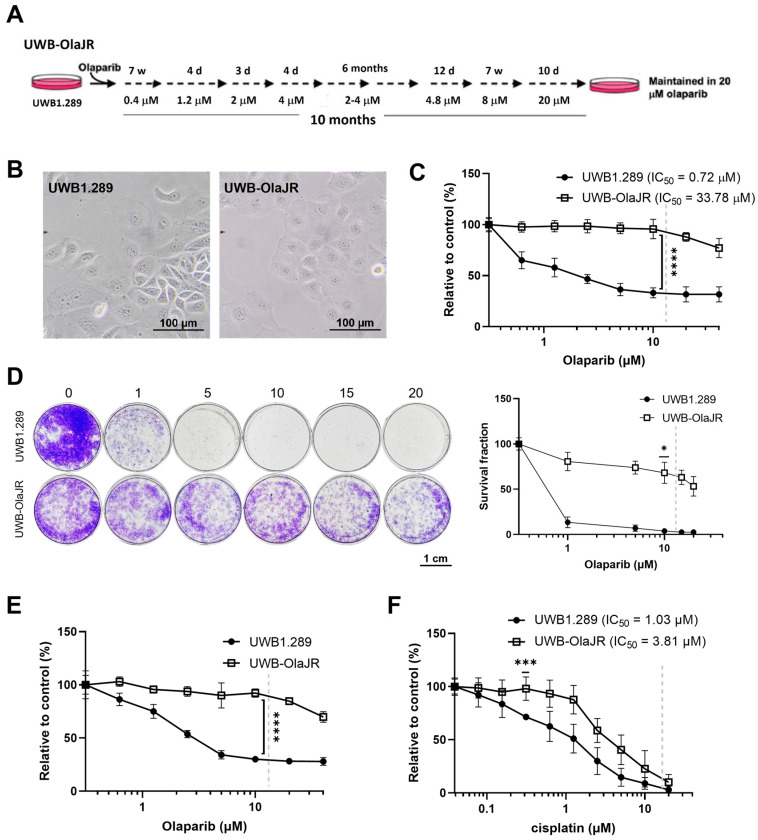
Establishment of the olaparib-resistant *BRCA1*m UWB1.289 cell line. (**A**) Schematic protocol for generating olaparib-resistant derivative of UWB1.289 (UWB-OlaJR) is shown. W, weeks; d, days. (**B**) Representative images showed similar cell morphology between parental UWB1.289 and olaparib-resistant UWB-OlaJR. (**C**) Cell growth was examined by XTT assay. UWB1.289 and UWB-OlaJR cells were treated with olaparib at indicated doses (0–40 µM) for 5 days. The value 0 was plotted as 0.31 µM for better visualization since 0 cannot be plotted on logarithmic scale. (**D**) Cell growth was examined by colony-forming assay. Cells were treated with olaparib at indicated doses for 10 days. (**E**) Olaparib-resistant UWB-OlaJR cell lines were maintained in the regular medium without olaparib for up to 8 weeks and subjected to cell growth assays. (**F**) Cells were treated with cisplatin at indicated doses for 5 days. Clinically attainable concentrations for each respective agent are denoted by the dotted line on each graph. All experiments were repeated in triplicate. Data are shown as mean ± SD. ****, *p* < 0.0001; ***, *p* < 0.001; *, *p* < 0.05.

**Figure 2 cells-13-01847-f002:**
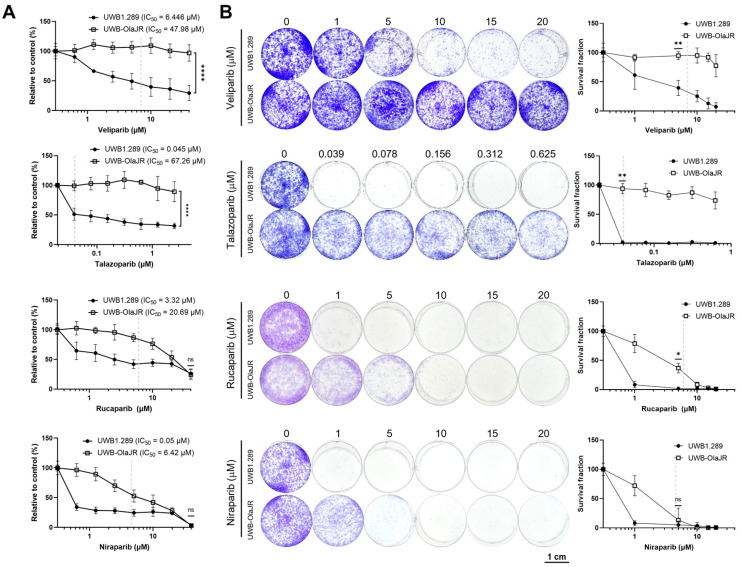
UWB-OlaJR shows varying degrees of resistance to different PARPis. (**A**) Cell growth was examined using the XTT assay. UWB-OlaJR cells were treated with veliparib, talazoparib, niraparib, and rucaparib at indicated doses for 5 days. The value 0 was plotted as 0.0195 µM in talazoparib and 0.3125 µM in veliparib, niraparib, and rucaparib for better visualization since 0 cannot be plotted on logarithmic scale. (**B**) Cell growth was examined by colony-forming assay. Cells were treated with olaparib at indicated doses for 10 days. The value 0 was plotted as 0.0195 µM in talazoparib for better visualization since 0 cannot be plotted on logarithmic scale. Clinically attainable concentrations for each respective agent are denoted by the dotted line on each graph. All experiments were repeated in triplicate. Data are shown as mean ± SD. ****, *p* < 0.0001; **, *p* < 0.01; *, *p* < 0.05; ns, not significant.

**Figure 3 cells-13-01847-f003:**
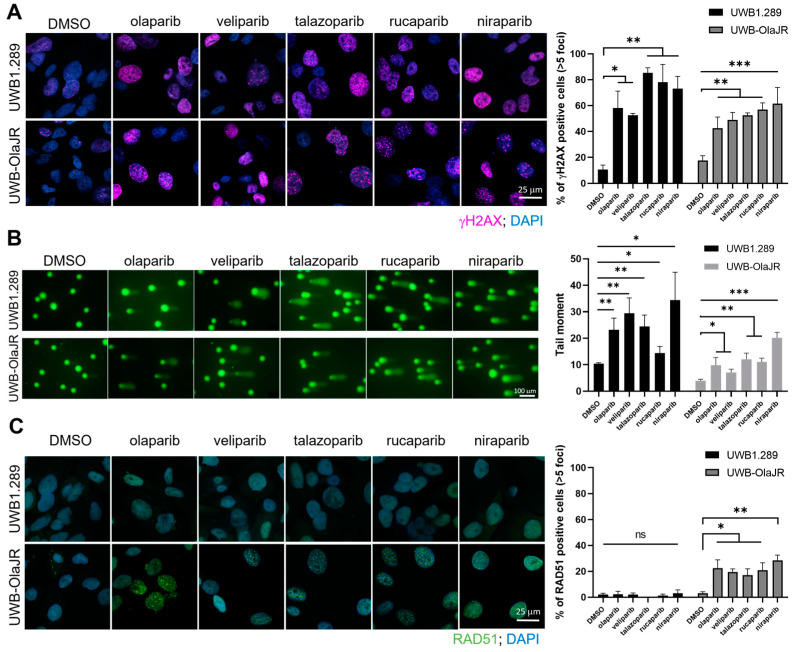
UWB-OlaJR shows varying degrees of resistance to different PARPis. Cells were treated with PARPis for 48 h, and the concentrations for each PARPi used are detailed in the Section 2. (**A**) Immunofluorescence staining of γH2AX foci was performed to examine the DSBs. Representative images were taken at ×60 magnification. Cells with >5 γH2AX foci were counted as γH2AX-positive cells. The percentage of γH2AX-positive cells is plotted. (**B**) Cells were treated with alternative PARPis, and its effect on DSBs was measured by neutral comet assay. Cell events were quantified using CometScore 2.0 software. The percentage of tail DNA is plotted. (**C**) Immunofluorescence staining of RAD51 foci was performed to examine the functionality of homologous recombination repair. Representative images were taken at ×60 magnification. Cells with >5 RAD51 foci were counted as RAD51-positive cells. The percentage of RAD51-positive cells is plotted. All experiments were repeated in triplicate. Data are shown as mean ± SD. ***, *p* < 0.001; **, *p* < 0.01; * *p* < 0.05; ns, not significant.

**Figure 4 cells-13-01847-f004:**
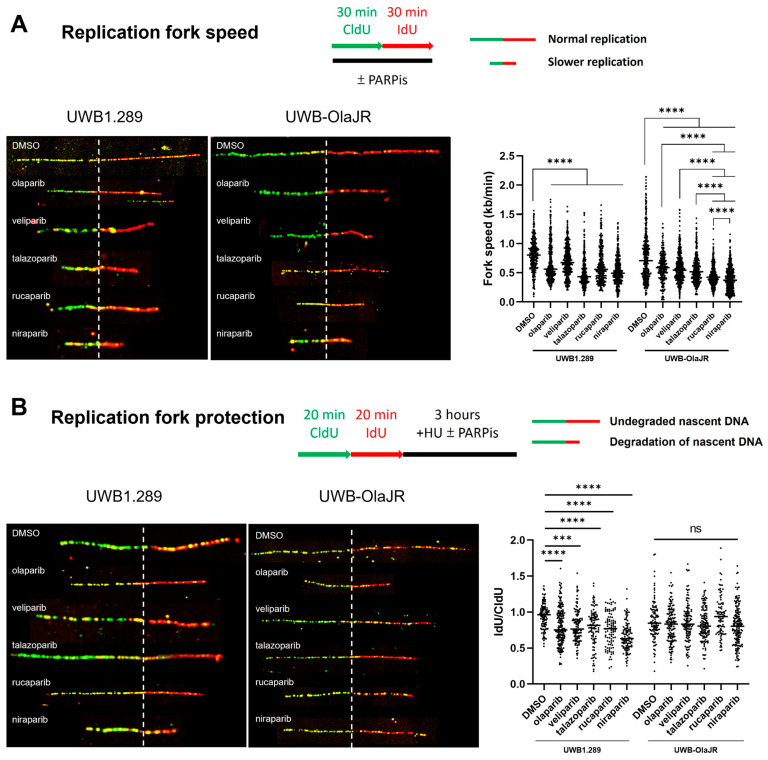
PARPis induce varying degrees of replication fork speed on UWB-OlaJR. DNA fiber assays were performed to study replication fork dynamics. The schematics protocol and DNA fiber images of representative strands are shown. Representative DNA fibers were visualized by immunostaining of CldU (green) and IdU (red). The concentrations for each PARPi used are detailed in the Section 2. (**A**) For measuring replication fork speed, cells were cultured with CldU followed by IdU for 30 min each with or without PARPis. Replicating 1 µm roughly corresponds to 2.59 kb. (**B**) For measuring replication fork protection, cells were cultured with CldU and IdU for 20 min, followed by treatment with 2 mM HU with or without PARPis. The ratio of IdU-labeled tracts to CldU was calculated to determine the degradation of nascent DNA. All experiments were repeated in triplicate. Data are shown as mean ± SD. ****, *p* < 0.0001; ***, *p* < 0.001; ns, not significant.

**Figure 5 cells-13-01847-f005:**
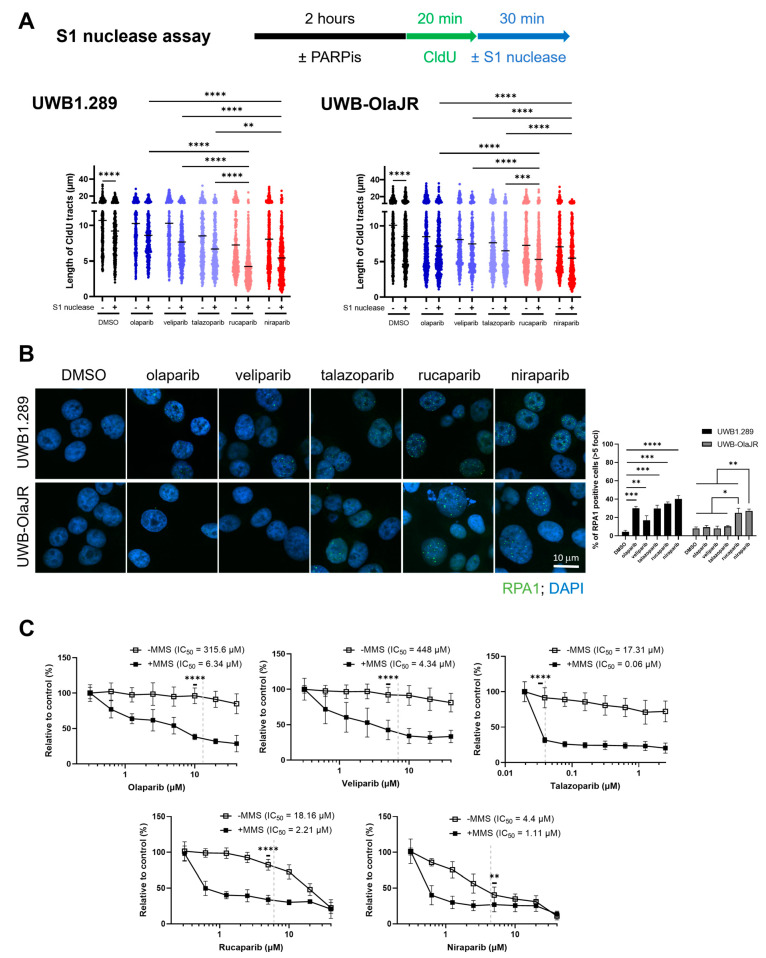
Resistance against olaparib, veliparib, and talazoparib can be rescued by SSB induction. (**A**) DNA fiber assays for CldU tracts with or without S1 nuclease incubation in indicated cells following PARPis treatment for 2 h. The concentrations for each PARPi used are detailed in the Methods. (**B**) Immunofluorescence staining of RPA1 foci was performed to examine the levels of ssDNA gaps. Representative images were taken at X60 magnification. Cells with >5 RPA1 foci were counted as RPA1-positive cells. The percentage of RPA1-positive cells is plotted. (**C**) Cell growth was examined using the XTT assay. UWB-OlaJR cells were incubated with 0.01% MMS for 10 min and washed away then, followed by treatment of olaparib, veliparib, talazoparib, niraparib, and rucaparib at indicated doses for 5 days. Clinically attainable concentrations for each respective agent are denoted by the dotted line on each graph. The value 0 was plotted as 0.0195 µM in talazoparib and 0.3125 µM in veliparib, niraparib, and rucaparib for better visualization, since 0 cannot be plotted on logarithmic scale. All experiments were repeated in triplicate. Data are shown as mean ± SD. ****, *p* < 0.0001; ***, *p* < 0.001; **, *p* < 0.01; *, *p* < 0.05.

**Table 1 cells-13-01847-t001:** Antibodies used for IF staining.

Antibodies	Dilution Fold	Catalog Number	Supplier
BRCA1	1:200	sc-6954 AF647	Santa Cruz Biotechnology (Dallas, TX, USA)
γH2AX	1:200	613408	Biolegend (San Diego, CA, USA)
RAD51	1:200	ab196449	Abcam (Cambridge, UK)
RPA1	1:200	199097	Abcam

**Table 2 cells-13-01847-t002:** Summary table of characteristics when UWB-OlaJR rechallenges with different PARPis.

UWB-OlaJR	Olaparib	Veliparib	Talazoparib	Rucaparib	Niraparib
Sensitivity	+	+	++	+++	+++
DSB induction	↑	↑	↑	↑	↑↑
RAD51 foci formation	+	+	+	+	++
RF speed	↓	↓	↓↓	↓↓↓	↓↓↓
RF protection	-	-	-	-	-
SSB induction	-	-	-	↑↑	↑↑↑
RPA1 foci	-	-	-	↑↑	↑↑↑

+, ++, +++: increasing levels of the indicated phenotype (e.g., sensitivity); -: no effect on the indicated phenotype; ↑: increasing levels of induction of the indicated phenotype; ↓: increasing levels of reduction in the indicated phenotype.

## Data Availability

The raw data supporting the conclusions of this article will be made available by the authors upon reasonable request.

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
