# Peer review of "Poly (ADP-Ribose) Polymerase Inhibitor Olaparib-Resistant BRCA1-Mutant Ovarian Cancer Cells Demonstrate Differential Sensitivity to PARP Inhibitor Rechallenge"

_cells, 2024, doi:10.3390/cells13221847_

Round 1

Reviewer 1 Report

Comments and Suggestions for Authors

The manuscript entitled: “PARP inhibitor olaparib-resistant BRCA1-mutant ovarian cancer cells demonstrate differential sensitivity to PARP inhibitor rechallenge” presented by Shih et al. is an interesting work exploring the Olaparib resistance in mutated OvCa cell lines and comparing its response with a broad range of PARP inhibitors to observe the cross-resistance. This paper is well prepared in terms of understanding the subject as well it is easy to read, which makes it available to the readers.

I’ve got a minor issue regarding the discussion. There is a lack of critical point of view of the authors about the limitations of the study and potential directions of future studies. Additionally, it could be beneficial to perform the in vivo studies, but due to the time limits, I understand that it will be impossible to do.

Author Response

Comment 1: There is a lack of critical point of view of the authors about the limitations of the study and potential directions of future studies.

Response 1: We have updated the Discussion to reflect the Reviewer’s comments as below.

Changes are shown in italics and bold (pages 14, lines 458-467): “There are limitations of this study. We acknowledge that additional olaparib-resistant BRCAm OvCa cell lines would help validate ssDNA gaps as a common resistance mechanism against PARPi in OvCa. Also, comparative analyses of ssDNA gap repair mechanisms and OFP pathway dynamics between UWB-OlaJR cells and other PARPi-resistant lines may uncover the compensatory DNA repair pathways for acquired resistance to PARPi. Moreover, the effects of other PARPis, specifically rucaparib and niraparib, on ssDNA gap formation and the backup OFP pathway requires further investigation in various PARPi-resistant OvCa preclinical models. Ultimately, clinical studies testing different PARPi reintroduction strategies will provide potential therapeutic implications in OvCa patients”.

Reviewer 2 Report

Comments and Suggestions for Authors

Authors investigated the differential sensitivity to different PARPis in olaparib-resistant BRCA1-mutant OvCa cell line (UWB-OlaJR) and found that UWB-OlaJR exhibited restored HR capability without BRCA1 reversion mutation or increased drug efflux. Moreover, UWB-OlaJR exhibited varying sensitivity to PARPis, showing cross-resistance to veliparib and talazoparib, but sensitivity with increased cytotoxicity to niraparib and rucaparib. Niraparib and rucaparib caused higher replication stress than others inducing greater DNA single-strand gaps compared to other PARPis, causing DNA damage and cell death. 

the manuscript is interesting and generally well written. Only some points could be improved. In particular:

Introduction: It deserves to be specified that also platinum-derived chemotherapeutics can be used for ovarian cancer treatment but chemoresistance occurence, mainly due to the increased antioxidant capacity of ovarian cance cells, is very frequent (see PMID: 38203758). This is an important point to introduce since it can further highlight the importance of the study done by the authors.

2.7. Immunofluorescence (IF) staining: I suggest to move the antibodies used in a dedicate table

Images size of colony assay are too small 

Authors must write the abbreviations in full length when mentioned for the first time

Author Response

Comment 1: Introduction: It deserves to be specified that also platinum-derived chemotherapeutics can be used for ovarian cancer treatment, but chemoresistance occurrence, mainly due to the increased antioxidant capacity of ovarian cancer cells, is very frequent (see PMID: 38203758). This is an important point to introduce since it can further highlight the importance of the study done by the authors.

Response 1: Thank you for your comments. We agree with the Reviewer that platinum-based chemotherapy and antioxidant-mediated chemoresistance are important aspects of ovarian cancer treatment. However, as PARP inhibitors (PARPis) resistance is an emerging clinical problem and our study specifically examines different mechanisms of action of PARPis in BRCA-mutant PARPi-resistant settings, we decided to maintain this focus on PARPi reintroduction strategies.

Comment 2: 2.7. Immunofluorescence (IF) staining: I suggest to move the antibodies used in a dedicate table.

Response 2: We have provided a new Table 1 as suggested. 

Comment 3: Images size of colony assay are too small.

Response 3: We have enlarged the colony images.

Comment 4: Authors must write the abbreviations in full length when mentioned for the first time.

Response 4: Thank you for the reminder.

Changes are shown in italics and bold. (Page 1, lines 32-35; Page 1-2, lines 42-43; Page 14, lines 451): “chromatin remodelers (nuclear receptor binding SET domain protein 2, amplification in liver cancer 1) [3-5], transcription factors (Krüppel-like factor 4) [6] and DNA damage repair proteins (breast cancer type 1 susceptibility protein (BRCA1), DNA polymerase β (POLβ), X-ray repair cross-complementing protein 1 (XRCC1)”, “PARP1 also protects reversed RFs by inhibiting RecQ protein-like 1 (RECQ1) against meiotic recombination 11 (MRE11)-mediated nucleolytic degradation [11, 12].”, “Furthermore, Cong et al. recently reported that loss of Fanconi anemia complementation group J (FANCJ) helicase…”

Reviewer 3 Report

Comments and Suggestions for Authors

In this study, the authors provide novel mechanistic insights into the reintroduction of niraparib and rucaparib for olaparib-resistant BRCA1m OvCa. The manuscript is straightforward, well written, and concise and has clear results. Definitely deserves to be published and is a valuable contribution to the “cells jounal. However, the following comments need to be addressed, as recommended.

[1]1. Introduction”, Page 1 of 17, Lines 34-37:

This poly ADP-ribosylation (PARylation) activity contributes to various DNA damage repair pathways, including single-strand break (SSB) repair processes, i.e., base excision repair and nucleotide excision repair, and double-strand break (DSB) repair response such as homologous recombination (HR) and non-homologous end joining [9].”.

This should be expanded with updated data. The authors should report that there have been six primary pathways of DNA damage repair deficiency identified, which are variably used to address DSB and SSB damage from a variety of mechanisms of injury. HR and nonhomologous end joining (NHEJ) recombination are the two major pathways responsible for repairing DSB. HR pathways become active in the S/G2 phase due to the availability of a sister chromatid, whereas NHEJ repairs DSB throughout all cell cycle phases except the M phase. NHEJ is faster than HR and mainly occurs in the G1 phase. Beyond the already-known proteins, such as Ku70/80, DNA-PKcs, Artemis, DNA pol λ/μ, DNA ligase IV-XRCC4, and XLF, new proteins are involved in the NHEJ, namely PAXX, MRI/CYREN, TARDBP of TDP-43, IFFO1, ERCC6L2, and RNase H2. Among them, MRI/CYREN has dual role, as it stimulates NHEJ in the G1 phase of the cell cycle, while it inhibits the pathway in the S and G2 phases.

Recommended reference: Boussios S, et al. BRCA Mutations in Ovarian and Prostate Cancer: Bench to Bedside. Cancers (Basel). 2022;14:3888.

[2] “4. Discussion”, Page 12 of 17, Lines 405-409:

“Currently, selection of one PARPi over another in the clinic is primarily based on the expected side effects profiles rather than efficacy, as all PARPis have shown similar overall response rates (ORR) and PFS in the upfront maintenance therapy setting for BRCAm OvCa patients [52].”.

At that point, the authors should mention that PARP–DNA complexes have the ability to interfere with DNA replication, and it has been indicated that PARP trapping is important for the cytotoxicity of PARP inhibitors. This explains the different magnitude of cytotoxicity exerted by different PARP inhibitors. Among PARP inhibitors that have already been evaluated, olaparib, niraparib, and rucaparib trap PARP approximately 100-fold more efficiently than veliparib.

Recommended reference: Ghose A, et al. Hereditary Ovarian Cancer: Towards a Cost-Effective Prevention Strategy. Int J Environ Res Public Health. 2022;19(19):12057.

Comments on the Quality of English Language

Minor editing of English language required.

Author Response

Comment 1: “Introduction”, Page 1 of 17, Lines 34-37:

“This poly ADP-ribosylation (PARylation) activity contributes to various DNA damage repair pathways, including single-strand break (SSB) repair processes, i.e., base excision repair and nucleotide excision repair, and double-strand break (DSB) repair response such as homologous recombination (HR) and non-homologous end joining [9].”.

This should be expanded with updated data. The authors should report that there have been six primary pathways of DNA damage repair deficiency identified, which are variably used to address DSB and SSB damage from a variety of mechanisms of injury. HR and nonhomologous end joining (NHEJ) recombination are the two major pathways responsible for repairing DSB. HR pathways become active in the S/G2 phase due to the availability of a sister chromatid, whereas NHEJ repairs DSB throughout all cell cycle phases except the M phase. NHEJ is faster than HR and mainly occurs in the G1 phase. Beyond the already-known proteins, such as Ku70/80, DNA-PKcs, Artemis, DNA pol λ/μ, DNA ligase IV-XRCC4, and XLF, new proteins are involved in the NHEJ, namely PAXX, MRI/CYREN, TARDBP of TDP-43, IFFO1, ERCC6L2, and RNase H2. Among them, MRI/CYREN has dual role, as it stimulates NHEJ in the G1 phase of the cell cycle, while it inhibits the pathway in the S and G2 phases.

Recommended reference: Boussios S, et al. BRCA Mutations in Ovarian and Prostate Cancer: Bench to Bedside. Cancers (Basel). 2022; 14:3888.

Response 1: We appreciate the reviewer’s comment.

We have revised the manuscript and updated the reference as follows [1] (changes are shown in italics and bold, page 1, lines 36-40): “Among the DNA damage repair pathways, poly ADP-ribosylation (PARylation) activity mainly contributes to single-strand break (SSB) repair processes, i.e., base excision repair and nucleotide excision repair, and double-strand break (DSB) repair response such as homologous recombination (HR), non-homologous end joining (NHEJ), and microhomology-mediated end joining (MMEJ) [9]”.

Comment 2: “Discussion”, Page 12 of 17, Lines 405-409:

“Currently, selection of one PARPi over another in the clinic is primarily based on the expected side effects profiles rather than efficacy, as all PARPis have shown similar overall response rates (ORR) and PFS in the upfront maintenance therapy setting for BRCAm OvCa patients [52].”.

At that point, the authors should mention that PARP–DNA complexes have the ability to interfere with DNA replication, and it has been indicated that PARP trapping is important for the cytotoxicity of PARP inhibitors. This explains the different magnitude of cytotoxicity exerted by different PARP inhibitors. Among PARP inhibitors that have already been evaluated, olaparib, niraparib, and rucaparib trap PARP approximately 100-fold more efficiently than veliparib.

Recommended reference: Ghose A, et al. Hereditary Ovarian Cancer: Towards a Cost-Effective Prevention Strategy. Int J Environ Res Public Health. 2022;19(19):12057.

Response 2: Thank you for this suggestion. We have already cited the references regarding the PARP trapping activity of PARPis as follows [2-4] (references 42, 43, lines 275-278; reference 63, lines 455-456).  

References

1          Ray Chaudhuri A, Hashimoto Y, Herrador R, Neelsen KJ, Fachinetti D, Bermejo R et al. Topoisomerase I poisoning results in PARP-mediated replication fork reversal. Nat Struct Mol Biol 2012; 19: 417-423.

2          Thomas A, Murai J, Pommier Y. The evolving landscape of predictive biomarkers of response to PARP inhibitors. J Clin Invest 2018; 128: 1727-1730.

3          Murai J, Pommier Y. PARP Trapping Beyond Homologous Recombination and Platinum Sensitivity in Cancers. Annu Rev Canc Biol 2019; 3: 131-150.

4          Cong K, MacGilvary N, Lee S, MacLeod SG, Calvo J, Peng M et al. FANCJ promotes PARP1 activity during DNA replication that is essential in BRCA1 deficient cells. Nat Commun 2024; 15: 2599.